# Methodology for Estimating the Spatial and Temporal Power Demand of Private Electric Vehicles for an Entire Urban Region Using Open Data

Florian Straub *[ID], Simon Streppel [ID] and Dietmar Göhlich [ID]

Chair for Methods of Product Development and Mechatronics, Technical University of Berlin,
Strasse des 17. Juni 135, 10623 Berlin, Germany; simon.streppel@campus.tu-berlin.de (S.S.);
dietmar.goehlich@tu-berlin.de (D.G.)
* Correspondence: florian.straub@tu-berlin.de; Tel.: +49-30-314-73895

**Abstract:** With continuous proliferation of private battery electric vehicles (BEVs) in urban regions, the demand for electrical energy and power is constantly increasing. Electrical grid infrastructure operators are facing the question of where and to what extent they need to expand their infrastructure in order to meet the additional demand. Therefore, the aim of this paper is to develop an activity-based mobility model that supports electrical grid operators in detecting and evaluating possible overloads within the electrical grid, deriving from the aforementioned electrification. We apply our model, which fully relies on open data, to the urban area of Berlin. In addition to a household travel survey, statistics on the population density, the degree of motorisation, and the household income in fine spatial resolution are key data sources for generation of the model. The results show that the spatial distribution of the BEV charging energy demand is highly heterogeneous. The demand per capita is higher in peripheral areas of the city, while the demand per m$^2$ area is higher in the inner city. For reference areas, we analysed the temporal distribution of the BEV charging power demand, by assuming that the vehicles are solely charged at their residential district. We show that the households' power demand peak in the evening coincide with the BEV power demand peak while the total power demand can increase up to 77.9%.

**Keywords:** electric vehicle; activity-based simulation; transportation electrification; charging power demand; spatial temporal distribution; open data

## 1. Introduction

Poor air quality in cities and continuously rising greenhouse gas emissions worldwide have led to a steady tightening of emission limits in recent years. The European Union has committed itself to reducing greenhouse gas emissions by 2050 by 80–95% of the level of 1990 [1]. In Germany, greenhouse gas emissions are planned to be reduced by 50% by 2030 compared to 1990 [2]. One key driver of this reduction is the substitution of internal combustion engine vehicles (ICEVs) with vehicles with alternative drive systems, primarily battery electric vehicles (BEVs). However, an increasing number of BEVs and the resulting large demand of electrical energy and power can lead to bottlenecks in the power supply if the necessary electrical grid infrastructure is not established or reinforced [3–6]. The investigations in this paper are restricted to the electrification of private internal combustion engine cars. The electrification of other urban vehicles such as buses or urban freight transport vehicles was investigated, for example, by the authors of [7–9]. The additional demand of private battery electric cars for electrical energy and power varies greatly depending on spatial and temporal factors. In general, areas with high population density and a high vehicle per capita rate have a higher demand for electrical energy compared to sparsely populated areas with few vehicles per inhabitant [6,10,11]. The authors of [3,4,6,12] show that the electrical energy and power demand for charging BEVs differs depending on the time of the day and the type of the day

(working day or weekend day). This large degree in variability creates difficulties concerning the planning of the expansion and optimisation of the electric grid, which is necessary to meet the additional charging energy and power demand. To overcome this limitation, data-based models with high spatial resolution are required in order to make realistic statements about the spatial and temporal energy and power requirements arising from the electrification of motorised individual traffic.

The authors of [13] developed a model based on diffusion theory that determines the spatial power requirements in the city of Porto, considering different charging capacities and five battery electric vehicle penetration rates from 10% to 100%. However, the study does not include a differentiated temporal determination of the power demand. By generating commuting travel chains for their investigation, the authors of [12] estimate the spatial and temporal distribution of the charging power demand of BEVs in urban areas. Result-relevant values such as the start time of the first trip or the covered trip distance are random and independent of the activity drawn from a normal distribution and a lognormal distribution, respectively. The distributions are adjusted to average values derived from a travel survey. The authors of [14] simulate the spatial and temporal distribution of the charging energy demand for an artificial city consisting of a city center, suburban areas, and connecting highways. All simulated persons and their vehicles undertake a round trip, which starts in a suburban area then goes to a random point within the city center and back. Within the city center, the persons decide whether to charge their car, depending on the depth of discharge. The arrival and departure times of the vehicles in the city center are drawn from a gamma distribution and are imprinted on the vehicles. The distribution is fitted to the results of a travel survey, which was conducted in Chicago.

Another approach to modelling the spatial and temporal energy and power demand deriving from electrification of the traffic is offered by activity-based models [10,11,15]. In activity-based models, the individual properties of individual persons within the research space are used to create full-day travel schedules for those persons. The resulting daily patterns for individual persons consist of a consecutive sequence of activities and trips. According to [15], activity-based models are particularly suitable for simulating the energy and power demand of BEVs. They capture the relationship between activity and mobility patterns, which can be used to determine the dwell times of the persons and their vehicles at different locations. The energy and power demand can then be determined by imposing charging strategies.

An example for an activity-based mobility model is the framework FEATHERS [16], which the authors of [11] used to simulate spatial and temporal charging power demand in the Flemish region in Belgium. Therefore, they divided the region into several districts. As there were no empirical data such as a travel survey available in the region, the authors used self-acquired data including "activity-travel diaries", information on "activity (re)scheduling decisions of individuals" and "data on household multi-day activity scheduling". The vehicle class category distribution, which has a significant impact on the energy and power demand (larger and heavier vehicles tend to have a higher energy consumption compared to smaller and lighter vehicles), matches the numbers for the entire region and not the individual districts. This results in a non-precise estimate of the spatial distribution of the energy and power demands. Furthermore, the authors do not verify their results by comparing them with the empirical data from the underlying survey. The authors of [10] used a household interview travel survey to create a activity-based mobility model for the Singapore urban area. After the division of Singapore into several districts, they used their model to estimate the spatial and temporal distributions of the charging energy demand for a working day and a weekend day, considering different electrification rates for private vehicles. Similar to [11], the vehicle class categories were drawn from a distribution that matches the overall vehicle class category distribution in Singapore. Additionally, the mobility model was not verified by comparison with the underlying survey results.

In their research on decarbonisation of the urban traffic, the authors of [17] used the activity-based simulation framework MATSim [18,19] to analyse the effects of complete replacement of the current population of ICEVs with BEVs in urban regions. The authors focused on electrification of the entire transport system including private vehicles, the public transportation sector, and commercial and municipal traffic. Since the authors studied the impact of ICEV electrification for the entire urban area of Berlin and not its districts, they did not focus on the spatial resolution of the occurring charging energy and power demand. The vehicle class categories were drawn from a distribution that matches the overall vehicle class category distribution in Berlin and not the distribution in the Berlin districts. Between the MATSim data basis, namely census data, and the resulting trips in the simulation, there are several process steps leading to a risk of inaccuracies in spatial resolution. For this reason, we decided to infer the activity pattern and the energy and power demand in the individual districts directly from the census data and other data sources.

The aim of this paper is to develop a model that supports electrical grid operators to detect and evaluate possible overloads within the electrical grid, deriving from the electrification of private ICEVs in urban regions. Therefore, we develop a methodology, the novelty of which lies in the generation of an activity-based mobility model through the direct combination of a survey on travel behaviour [20–22], with data sets containing information about the population density [23], the degree of motorisation (which indicates the amount of vehicles per 1000 inhabitants) [24], and the household income in fine spatial resolution [25,26]. Through this direct combination, our model provides data-based results with high spatial and temporal resolution. Furthermore, the fine resolution of the household income enables us to determine the vehicle class category distribution in fine spatial resolution, which has, as we discussed in the previous paragraphs, a significant impact on the energy and power demand and is neglected by other researchers. In contrast to most other scientific works, we do not use a person-based but a vehicle-based approach. This allows us to realistically depict the use of the same vehicle by several persons (for example, in family groups). Since all used data sets are openly available and therefore relatively easy to acquire in most places of the world, our methodology is transferable to other urban areas. We apply our methodology to the urban area of Berlin and assume a scenario where 100% of the current private car population is electrified. We consider a working day and a Saturday in our simulation. We exclude Sundays from our simulation as 69% of the vehicles remain parked on Sundays, compared to 56% on Saturdays. In addition, the average distance of all vehicles is 23.9 km on Sundays, which is below the 26.1 km on Saturdays [22]. A higher energy and power demand on Sundays compared to Saturdays is therefore not to be expected. By applying the charging strategy "home-charging", we calculate the spatial and temporal energy and power demand distribution resulting from charging the BEVs.

In the city of Berlin, 40% of the population has access to private car parking spaces [22] while the remaining cars are parked in public spaces. Therefore, in this paper "home-charging" does not mean that all cars are charged in private parking lots at the owner's place of residence but that they are charged within the district where the owner of the vehicle lives. We assume that sufficient public charging infrastructure is available. We chose the strategy "home-charging" since, according to [27], 65% of the German population prefers home charging to charging at public charging stations (15%) or at work (7.5%). Reference [27] is a study on the acceptance of e-mobility in Germany. It was conducted in 2019 by interviewing 1200 German households.

This paper is structured as followed: In Section 2, our methodology is described and the implemented scenario is presented. The results are shown and analysed in Section 3. Finally, the main conclusions of this paper are derived in Section 4.

## 2. Methodology

The methodology used for our mobility model consists of three main parts. As a first step, we divide the city of Berlin into districts and determine for each the number of electric vehicles and the vehicle class categories within it. To do so, we use data sets containing

information about the population density [23], the degree of motorisation [24], and the household income within the districts [25,26]. In the second step, we generate individual daily patterns for the vehicles based on a household travel survey [20–22]. Finally, we use these daily patterns to determine the energy and power demand generated by charging the BEVs for a working day and a Saturday. These results are compared to the base power demand of the households within the districts.

### 2.1. District Classification and Vehicle Population

Most activity-based mobility models share one disadvantage. When the daily patterns of the persons are created and the means of transport is chosen, they assign each person their own vehicle. Therefore, multiple use of one vehicle is not considered. Since the dwell times of the vehicles are then determined incorrectly, the determination of the temporal distribution of the charging power demand is inaccurate. To overcome this problem, in our mobility model, we do not consider the individual persons but the vehicles within the system boundaries. We assume a scenario where 100% of the current private car population is electrified having the same share regarding vehicle class category as the current car population. Due to the lack of open data, commercially used vehicles such as delivery vehicles or taxis are not considered in our study. According to the "Kraftfahrtbundesamt", which is the German Federal Motor Transport Authority, 1.203 million cars were registered in Berlin in 2018, of which 1.045 million of them were private cars [28]. In order to distribute this number of vehicles spatially correctly, we make use of the official classification of the Berlin administration [29], which divides the twelve Berlin districts into 448 subdistricts, called "Lebensweltlich orientierte Räume" (LOR). The LOR classification was firstly introduced in 2006. Within each LOR, the structure of the contained buildings and the socioeconomical status of the inhabitants are similar. The LORs are usually separated from each other by major roads, rivers, or rails.

We derive the number of inhabitants for each LOR from [23], which is a freely available statistic provided by the Berlin authorities. The statistic is created by counting all registered residents within the LORs in 2018. The LOR classification, the population density, and the number of inhabitants within the LORs are depicted in Figure 1.

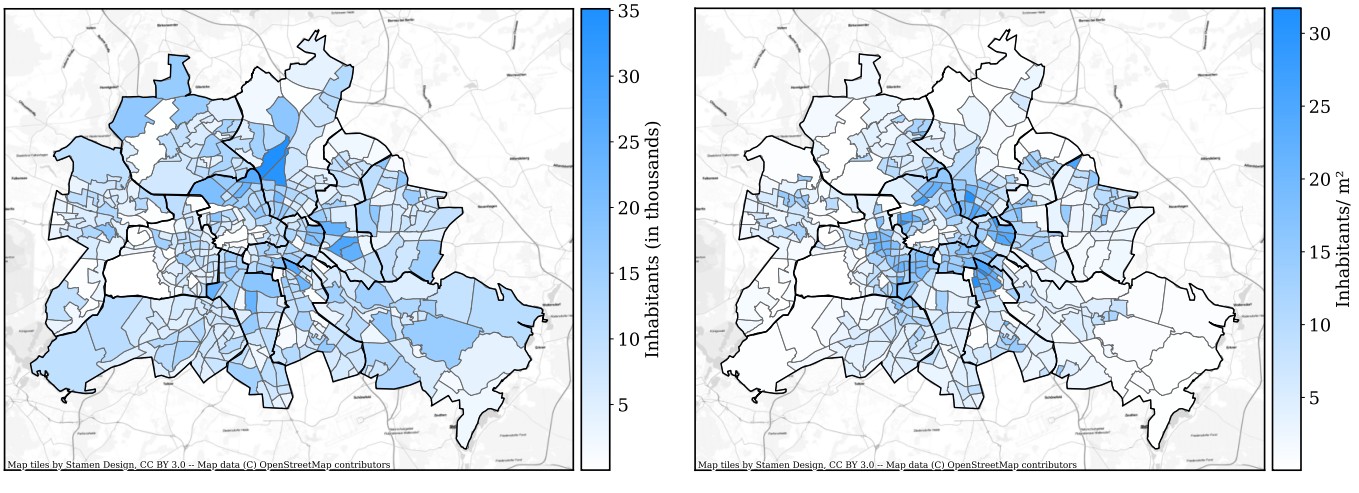

(**a**) Inhabitants in the Berlin LORs　　　　　(**b**) Population density in the Berlin LORs
**Figure 1.** Distribution of the population in the Berlin subdistricts, "Lebensweltlich orientierte Räume" (LORs).

To estimate the number of vehicles in each LOR, we use freely available data about the degree of motorisation (amount of vehicles per 1000 inhabitants) in each LOR [24]. The degree of motorisation for the Berlin LORs is depicted in Figure 2. It can be seen that the degree of motorisation is higher in LORs that are located in the peripheral areas of the city compared to the city center. By multiplying the amount of vehicles per person with the

total amount of persons within the LORs, we calculate the number of vehicles for each LOR separately.

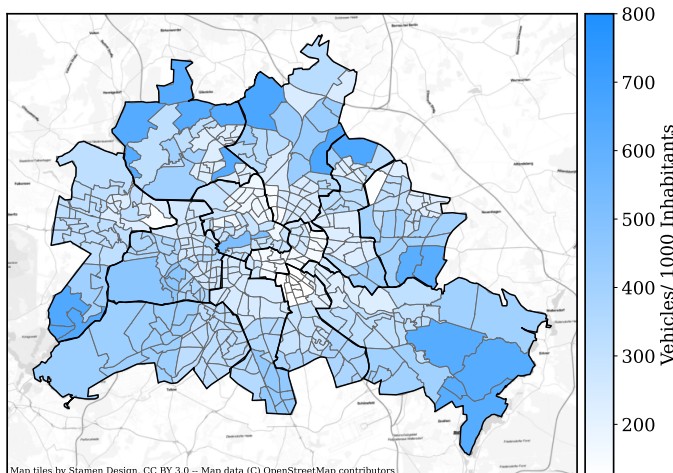

**Figure 2.** Degree of motorisation in the Berlin LORs.

Since the BEVs' energy consumption is highly dependent on the vehicle class category (larger and heavier vehicles tend to have a higher energy consumption compared to smaller and lighter vehicles), the vehicle class category distribution for all vehicles within the LORs has to be determined. To do so, we make use of the 2017 German household travel survey (GHTS) [20–22]. The survey contains information on 316,361 persons from 156,420 households in Germany and on almost one million trips. For the study, the members of a household were asked about their activities and trips during reference days. This allowed for conclusions on daily travel patterns. Furthermore, as a classical household travel survey, the study contains sociodemographic information such as gender and age, or the income of the household. It contains further information, such as the amount of vehicles in the household and information on the vehicle class category. The data set can be obtained freely, either as raw data [22] or already pre-evaluated [21]. As the data set contains information on the travel behaviour in all of Germany, we initially limited the data set to households and persons within the city of Berlin. Furthermore, we excluded all trips that were not undertaken by car.

According to the GHTS data set, the amount of cars per household in Berlin correlates with the income of the household. In general, households with higher income own more vehicles compared to households with lower income. Furthermore, the GHTS indicates that the vehicle class category distribution in Berlin also depends on the income of the household. Households with high income tend to own larger vehicles. The distributions are depicted in Table 1. In the GHTS data set, the household's income is divided into five income categories. For a one-person household, the average net income for the income category "very low" lies below €900, between €900 and €1500 for the category "low", between €1500 and €2800 for the category "medium", between €2800 and €4000 for the category "high", and above €4000 for the category "very high".

**Table 1.** Correlation between household income and vehicles in Berlin.

| Household Income | Average Amount of Cars per Household | Relative Frequency Vehicle Class Category | | | |
|---|---|---|---|---|---|
| | | **Mini Compact** | **Compact** | **Medium** | **Large** |
| very low | 0.3 | 0.31 | 0.37 | 0.27 | 0.054 |
| low | 0.5 | 0.35 | 0.35 | 0.25 | 0.052 |
| medium | 0.7 | 0.27 | 0.39 | 0.26 | 0.074 |
| high | 1.0 | 0.24 | 0.34 | 0.32 | 0.11 |
| very high | 1.3 | 0.18 | 0.30 | 0.34 | 0.19 |

In their research on the socio-structural situation in Berlin, the authors of [30] show that the residential area quality strongly correlates with the household income. This means that households with higher income tend to live in areas with higher quality. High-quality residential areas are mostly located close to the city center and are usually characterised by rather high greening. In comparison, low-quality residential areas mostly have high building density, which may be intermixed with or adjacent to commerce and industry. In addition, simple residential areas usually have little greenery [26]. According to [30], we estimate the shares of household income in the LORs by analysing the residential area quality [26]. To verify our results, we check whether our results match the distributions of the household income in the twelve Berlin districts. We derived those distributions from the 2018 census data [25]. The census is conducted on a yearly basis in Germany by interviewing 1% of the population. It is freely available and provides information on the economic and social situation of the population in Germany, such as information on household and family structures, employment, or income situation. The resulting distribution of the household income for the Berlin LORs is depicted in Figure 3.

By combining the aforementioned information, we obtain the amount of vehicles for each vehicle class category within the LORs.

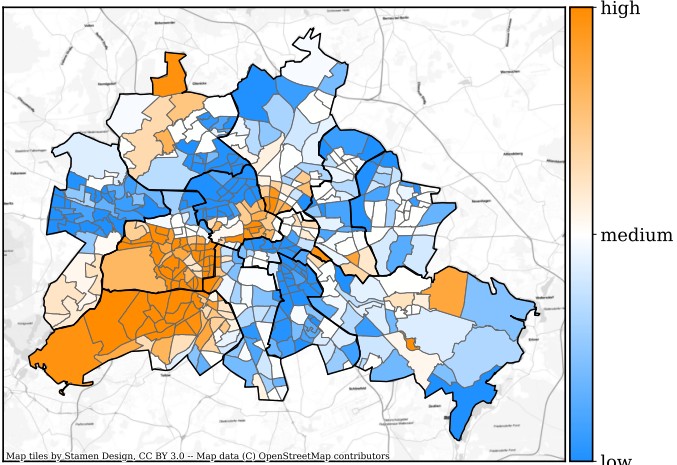

**Figure 3.** Household income distribution in the Berlin LORs.

*2.2. Mobility Profiles*

As a basis for the mobility profiles of the vehicles, we analysed the data set from the 2017 German household travel survey (GHTS) [20–22]. A mobility profile is a sequence of activities and trips between those activities. For our study, we assumed that BEV drivers show the same mobility behaviour as ICEV users. In Figure 4, a general mobility profile of a vehicle is shown. The vehicle starts at "Home" in the morning before spending 7 h and 50 min at the activity "Working" and 30 min at the activity "Shopping". It arrives back "Home" at 17:00. In order to create mobility profiles, we extracted the relevant information on trips (activity, departure and arrival times, and trip distance) from the GHTS data. The data set distinguishes more than 25 activities. Similar to [10,19], we divide those activities into five superordinate activities for simplification reasons: "Working", "Shopping", "Education", "Home", and "Else".

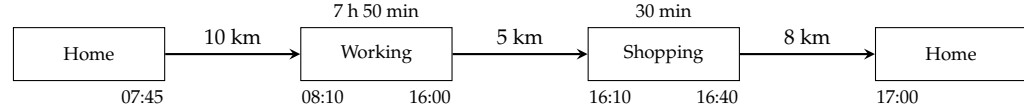

**Figure 4.** Example of a general mobility profile.

The GHTS data set is evaluated for an average working day (Monday–Thursday) and Saturdays. We exclude Fridays from our analysis, as the average distance of all vehicles is

15.1 km on Fridays, which is less than half of the 32.6 km average distance on the other working days. On Fridays, 62% of the vehicles remain parked, compared to 45% on the other working days. Sundays are excluded from our analysis, as 69% of the vehicles remain parked compared to 56% on Saturdays. In addition, the average distance of all vehicles is 23.9 km on Sundays, which is below the 26.1 km on Saturdays. A higher energy and power requirement on Sundays compared to Saturdays and on Fridays compared to the other working days is therefore not to be expected. The creation of the mobility profiles is performed successively for each individual vehicle within the individual LOR. The basis for this process is the distribution of the vehicle class categories we derive in Section 2.1. For each of the four vehicle class categories, we considered three reference vehicles, making in total a sum of twelve considered reference vehicles, which are depicted in Table 2. For each reference vehicle, the energy demand per 100 km and the vehicles' battery capacity can be obtained from Table 2. The average consumption is divided into inner-city trips, which are characterised by distances of less than 20 km and outer-city trips. The values of the energy consumption and the battery capacity are based on test drives of the "Allgemeine Deutsche Automobil Club" (ADAC), a German motoring association, and already include charging losses [31].

**Table 2.** Reference vehicles.

| Class | Model | Battery Capacity (kWh) | Inner City Consumption (kWh/100 km) | Outer City Consumption (kWh/100 km) |
|---|---|---|---|---|
| Mini compact | Mitsubishi i-MiEV [32] | 15.9 | 11.3 | 16.9 |
| | Renault Zoe [33] | 64.3 | 14.5 | 19.0 |
| | VW e-Up! [34] | 18.6 | 14.0 | 17.7 |
| Compact | BMW i3 [35] | 48.8 | 13.0 | 17.9 |
| | Hyundai Kona E [36] | 73.9 | 14.0 | 19.5 |
| | VW e-Golf [37] | 34.9 | 12.7 | 18.2 |
| Medium | Kia e-Niro [38] | 72.3 | 12.5 | 18.1 |
| | Nissan Leaf [39] | 68.4 | 17.2 | 22.7 |
| | Tesla Model 3 [40] | 60.0 | 17.4 | 19.3 |
| Large | Audi e-tron [41] | 94.3 | 23.5 | 25.8 |
| | Mercedes EQC [42] | 93.1 | 23.0 | 27.6 |
| | Tesla Model S [43] | 100.4 | 21.2 | 24.2 |

The general process for creating the mobility profile for one vehicle consists of six main steps (1)–(6). The process is depicted in Figure 5 and is described below. Taking the vehicle class category of the vehicle as input, one of the three reference vehicles is drawn from Table 2 with equal distribution (1).

In (2), the number of trips per day for the vehicle is drawn. We derive the underlying discrete probability distribution by analysing the GHTS data set. A separate probability function is evaluated for each vehicle class category. If the number of trips is zero, no mobility profile is created for the vehicle and the process starts with the next vehicle. If the number of trips is greater than zero, the starting time of the first trip is drawn from a probability distribution we generated for a 24 h interval by analysing the GHTS data set (3). The probability distributions we use in (3) are depicted in Figure 6 for a working day (Monday–Thursday) and a Saturday. On a working day, one sharp peak can be seen at around 08:00 in the morning. This is due to the fact that many vehicles are used early in the morning on the way to work. In comparison, two less sharp peaks can be seen on Saturdays. Vehicles tend to be used on weekends for shopping or leisure activities, which start either in the late morning or afternoon.

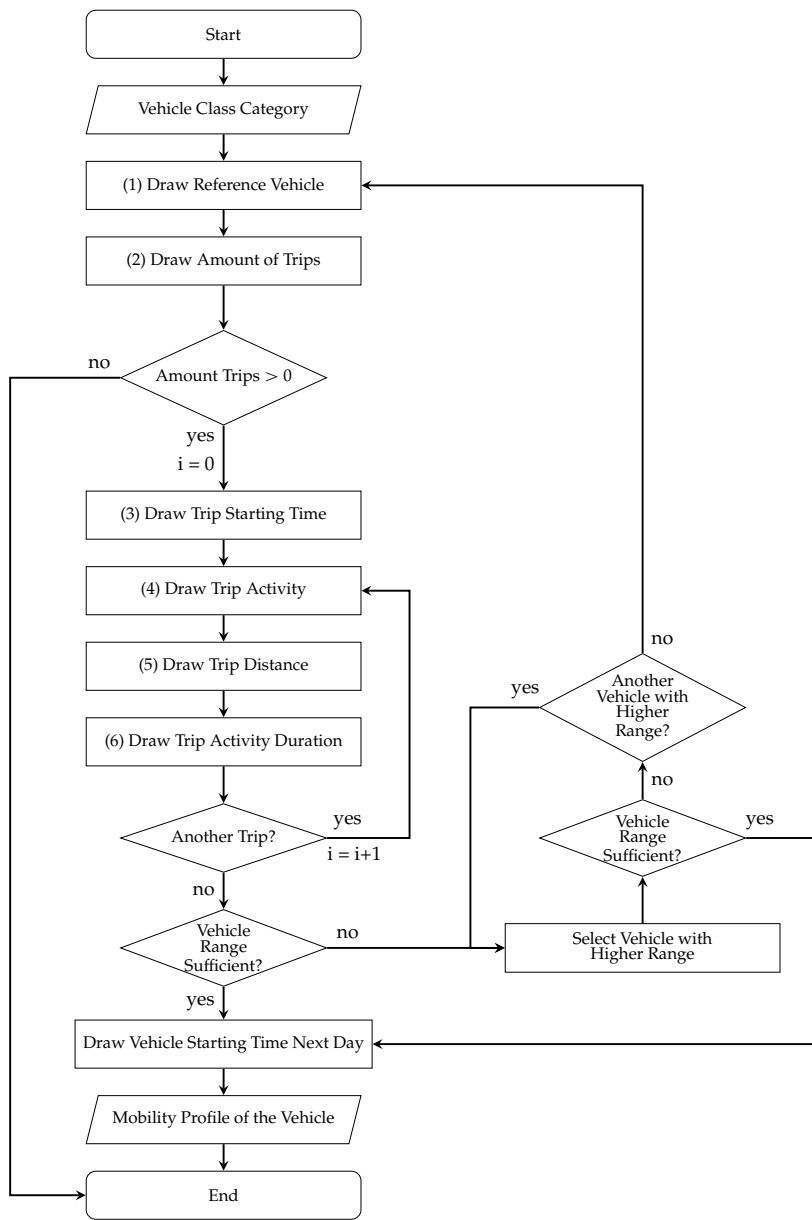

**Figure 5.** General process for creating the mobility profile for one vehicle.

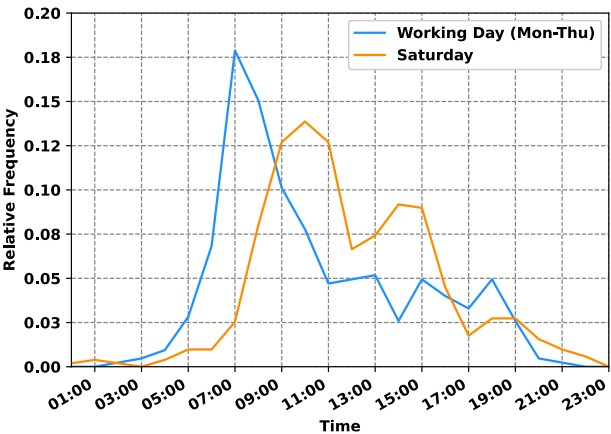

**Figure 6.** Starting time of the first trip on a working day (Monday–Thursday) and a Saturday.

Depending on the trip's starting time, the trip's activity is drawn afterwards (4). For this purpose, we use discrete distribution functions on an hourly basis, which provide information about the probability that a certain activity will occur.

In (5), the distance of the trip is drawn depending on the trip's activity. For the activities "Working", "Shopping", and "Home", the underlying probability distributions are depicted in blue in Figure 7a–c for a working day and in Figure 7d–f for a Saturday. It can be seen that "Working" trips on working days usually cover higher distances compared to "Shopping" and "Home" trips. The trip distance for 80% of the "Shopping" trips is below 11 km, while it is below 16 km for "Home" trips and 29 km for "Working" trips. On Saturdays, the trip distance for 80% of the "Shopping", "Working", and "Home" trips is about 19 km. The distributions in (4) and (5) are generated by analysing the GHTS data set.

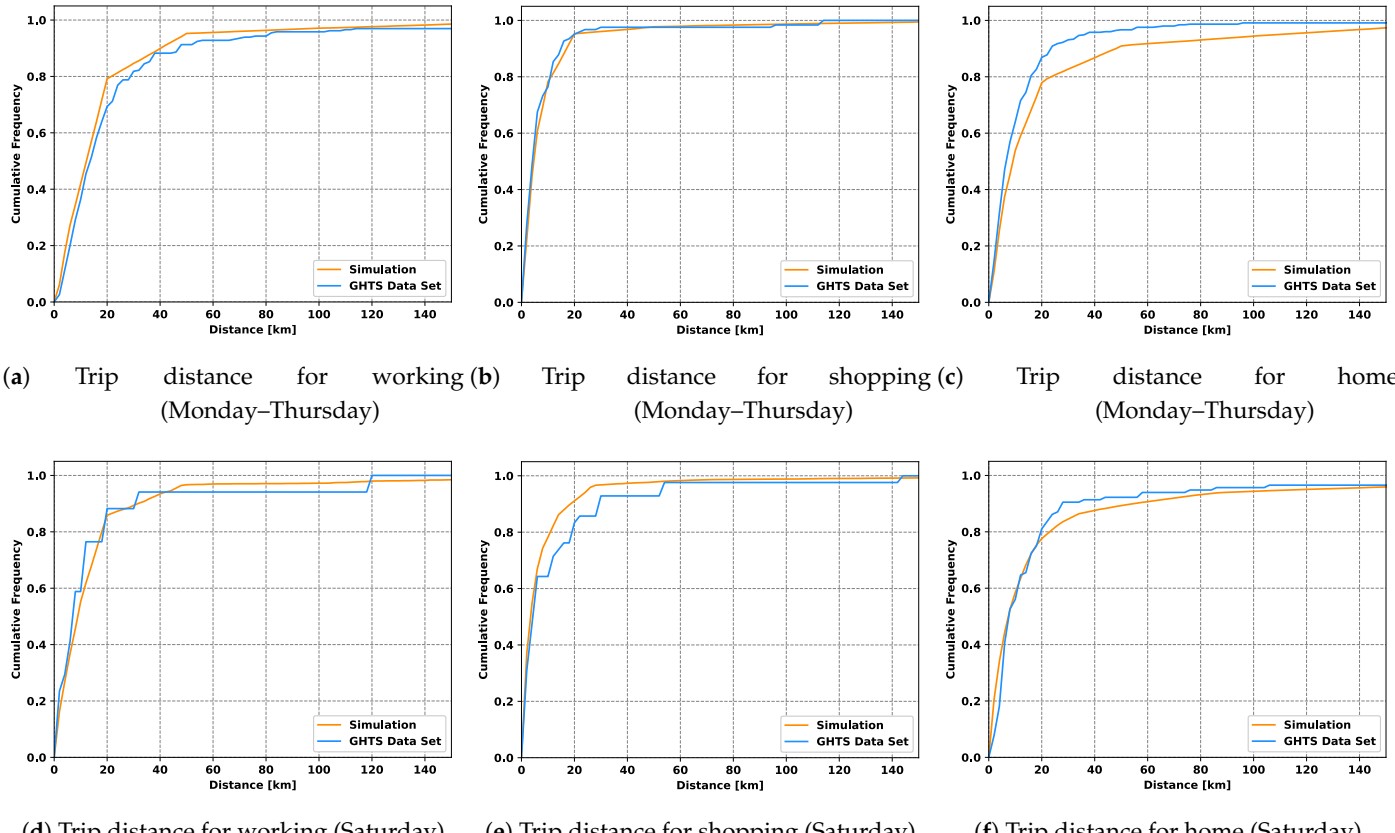

(**a**) Trip distance for working (Monday–Thursday)

(**b**) Trip distance for shopping (Monday–Thursday)

(**c**) Trip distance for home (Monday–Thursday)

(**d**) Trip distance for working (Saturday)

(**e**) Trip distance for shopping (Saturday)

(**f**) Trip distance for home (Saturday)

**Figure 7.** Cumulative distribution function for trip distance—simulation and German household travel survey (GHTS) data set comparison.

To determine the respective travel times for the trips, we use speed assumptions. We use [44] to estimate the activity duration in (6). Reference [44] is a survey from German authorities on the time usage behaviour of German citizens. The statistic provides information on how much time is spent on different activities such as working, shopping, education, or leisure activities. The survey was conducted in 2012, the third time after 1992 and 2002. For the survey, a total of 11,000 persons in approximately 5000 households in Germany were interviewed.

If the vehicle undertakes further trips, the loop is repeated with (4). The starting time of the next trip is the ending time of the last trip. If the last trip is for the activity "Home", the loop is repeated with (3) and a new starting time is drawn.

Finally we check whether the total driven distance between two consecutive charging events is covered by the range of the reference vehicle. As we only consider "home-charging", a charging event can only occur if the activity is "Home". If the vehicle's range is not sufficient, a vehicle with a higher range within the same vehicle class category is

drawn. If the vehicle's range is still too small to cover the driven distance, a vehicle with a sufficient range is searched for in the other vehicle class categories. If the total driven distance between two consecutive charging events exceeds the maximum range of all reference vehicles, the vehicle is re-simulated starting with (1). If the total driven distance can be covered by the reference vehicle, steps (2) and (3) are repeated to determine whether and when the vehicle drives off again the next day.

*2.3. Assumptions to Calculate the Spatial and Temporal Power Demand*

Based on the created mobility profiles for all privately owned vehicles in Berlin, we simulated 24 h of a working day (Monday–Thursday) and a Saturday to derive the spatial and temporal distribution of the BEV charging energy and power demand. Vehicles that started within this period are allowed to return later than 24:00. The results are then projected onto a 24 h interval. We made the following assumptions for the simulation in this paper:

- The vehicle's state of charge (SOC) at the start of the day is 100%.
- Charging exclusively at the home LOR. Sufficient public charging infrastructure is available.
- Charging starts immediately upon arrival at home and does not end until the vehicle is fully charged or drives off again.
- Constant charging power, which is independent of the SOC. To show the effects of different charging powers on the spatial and temporal distribution of the charging power demand, we run our simulation for charging powers of 3.7 kW and 11 kW, respectively.
- Power demand can be fully covered by the electrical grid at any time.

## 3. Results and Discussion

The first part of this section compares the simulation results with the GHTS data set. The comparison results are shown and discussed for the vehicles' trip distance distribution, the vehicles' average daily distances, and the amount of moving vehicles in Section 3.1; for the vehicle class category distribution in Section 3.2; and for the vehicles' trip starting time distribution in Section 3.3.

In the second part of this section, we show and discuss in Section 3.4 the simulated result of the spatial distribution of the charging energy demand. In Section 3.5, the results of the temporal distribution of the charging power demand are presented and discussed.

*3.1. Trip Distance and Moving Vehicles—Simulation and GHTS Data Comparison*

In Figure 7, a comparison of the cumulative distribution function of the trip distance of the simulation (blue) and the GHTS data set (orange) is shown. Figure 7a–c show the results for a working day (Monday–Thursday), Figure 7d–f show the results for Saturday. The figures are depicted for the activities "Working", "Shopping", and "Home". The steps in the GHTS data graphs result from the self-reported travel times in the survey. Participants tend to report rounded numbers rather than give exact values. As it can be seen, the general distribution shape matches for all activities and days. In Figure 8, the modelling errors of the results in Figure 7 are depicted as a boxplot. The blue square indicates the mean error. The boxplot's whiskers are modeled in such way that they represent the 2.5% quantile and the 97.5% quantile, respectively; 95% of all errors are thus within the whisker limits. The mean error is between −6.1% and 2.6% for a working day and between −2.0% and 3.3% on a Saturday. This indicates that the simulation represents the GHTS data with high accuracy.

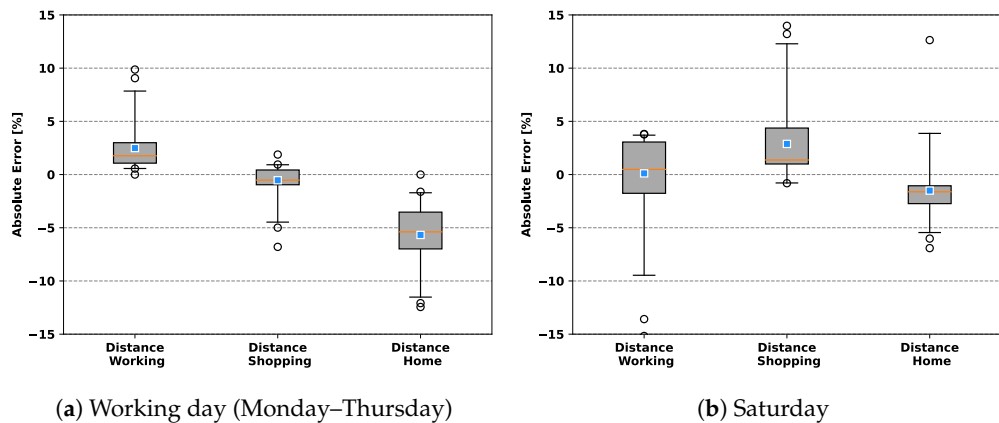

(**a**) Working day (Monday–Thursday)          (**b**) Saturday

**Figure 8.** Modelling error distance distributions. The blue square depicts the mean error.

Table 3 compares the average daily distance driven by the simulated vehicles with the GHTS data set. The relative error is −10.4% for working days (Monday–Thursday) and −5.7% on Saturdays. This error can be easily explained. As we show in Section 2.2, the daily distances of the simulated vehicles are limited. Vehicles that cover large daily distances are not represented in our simulation.

For working days (Monday–Thursday) trips with high distances are mostly "Working" trips, as can be seen in Figure 7a. The GHTS data set indicates that 95% of all "Working" trips are below 85 km, whereas the remaining 5% can reach values up to 750 km. On Saturdays, trips with high distances are mostly "Home" trips, as can be seen in Figure 7f. The GHTS data set shows that 95% of all "Home" trips on Saturdays are below 80 km whereas the remaining 5% can reach values up to 320 km. "Home" trips on Saturdays are higher compared to working days, as persons tend to travel higher distances for their leisure activities on Saturdays, which results in accordingly higher home distances.

This small share of trips with high distances in the GHTS data set lead to a higher average daily distance compared to our simulation. As time-consuming, inter-trip charging is necessary to cover high-distance trips, we assume that those trips are not undertaken with a private BEV but with other vehicles (e.g., shared vehicles) with alternative drives (e.g., hydrogen) in the future. Therefore, the reliability of the model is not affected by this error.

In Table 3, the proportion of moving vehicles for the simulation and the GHTS data set is shown. The relative error is 0.18% on a working day and 1.15% on a Saturday, which indicates that the simulation represents the GHTS data set with high accuracy.

**Table 3.** Comparison of the simulation and the GHTS data set—average daily distance and moving vehicles.

|  | Type of Day | GHTS Data Set | Simulation | Relative Error |
|---|---|---|---|---|
| Average Daily Distance | Working Day | 32.6 km | 29.2 km | −10.4% |
|  | Saturday | 26.1 km | 24.6 km | −5.7% |
| Percentage Moving Vehicles | Working Day | 54.8% | 54.9% | 0.18% |
|  | Saturday | 43.3% | 43.8% | 1.15% |

### 3.2. Vehicle Class Category Distribution—Simulation and GHTS Data Comparison

In Table 4, the simulated vehicle class category distribution is compared to the GHTS data set. The relative error is 5.9% for the mini compact class, −1.7% for the compact class, −1.0% for the medium class, and −6.3% for the large class; hence, our simulation represents the data set well.

**Table 4.** Comparison of the simulation and the GHTS data set—vehicle class category distribution.

| Vehicle Class Category | GHTS Data Set | Simulation | Relative Error |
|---|---|---|---|
| Mini Compact | 25.5% | 27.0% | 5.9% |
| Compact | 36.2% | 35.6% | −1.7% |
| Medium | 28.7% | 28.4% | −1.0% |
| Large | 9.6% | 9.0% | −6.3% |

### 3.3. Trip Starting Time—Simulation and GHTS Data Comparison

In Figure 9, a comparison of the simulated cumulative distribution function of the trip's starting time (blue) and the cumulative distribution function derived from the GHTS data set (orange) is shown. Figure 9a–c show the results for a working day (Monday–Thursday), Figure 9d–f show the results for Saturday. The figures are depicted for the activities "Working", "Shopping", and "Home". While the general distribution shape for the activity "Home" matches well, errors exist for the activities "Working" and "Shopping". Further improving the fit of those distributions would be a valuable future improvement. In Figure 10, the modelling errors of the results in Figure 9 are depicted as a boxplot. The mean error is between −4.3% and 2.6% for a working day (Monday–Thursday) and between −5.3% and 3.1% on a Saturday.

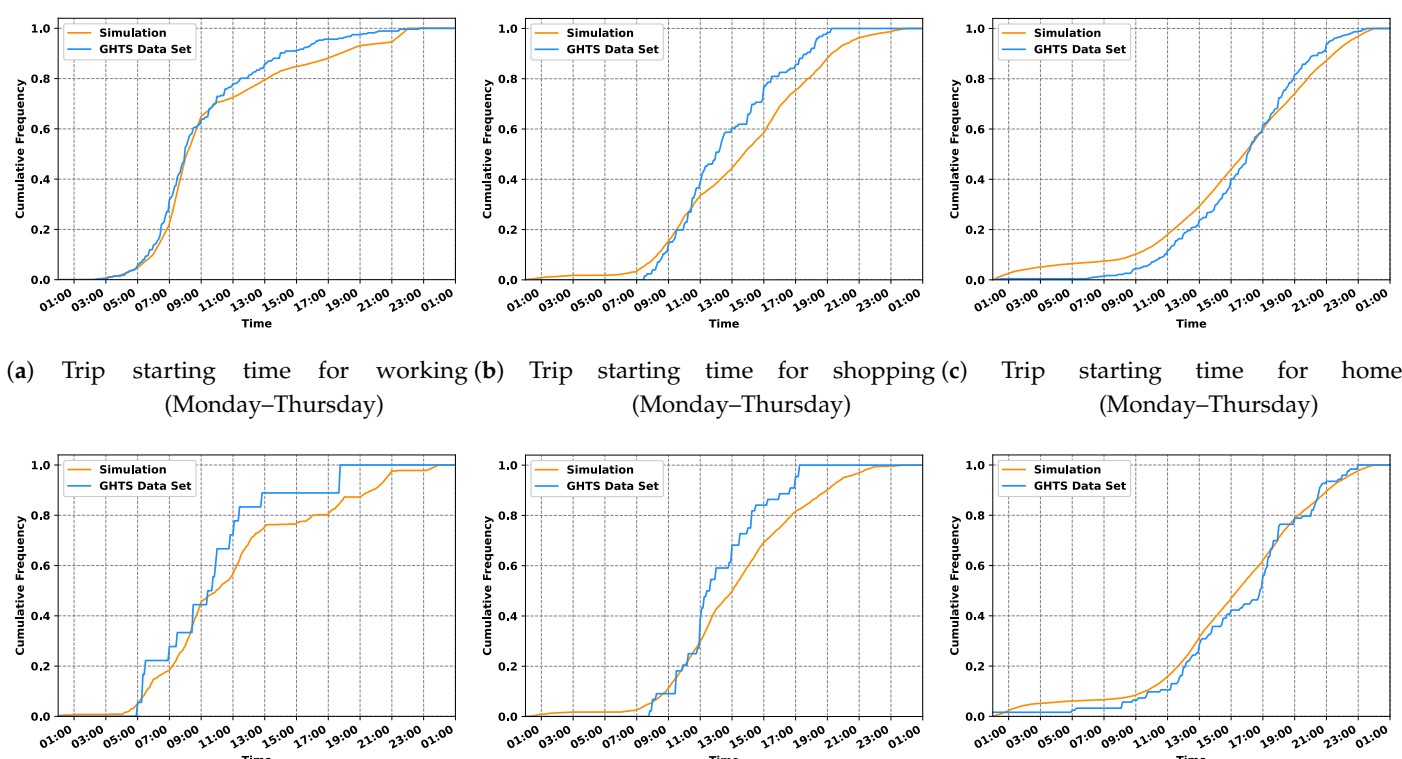

(**a**) Trip starting time for working (Monday–Thursday)

(**b**) Trip starting time for shopping (Monday–Thursday)

(**c**) Trip starting time for home (Monday–Thursday)

(**d**) Trip starting time for working (Saturday) (**e**) Trip starting time for shopping (Saturday) (**f**) Trip starting time for home (Saturday)

**Figure 9.** Cumulative distribution function for trip starting time—simulation and GHTS data set comparison.

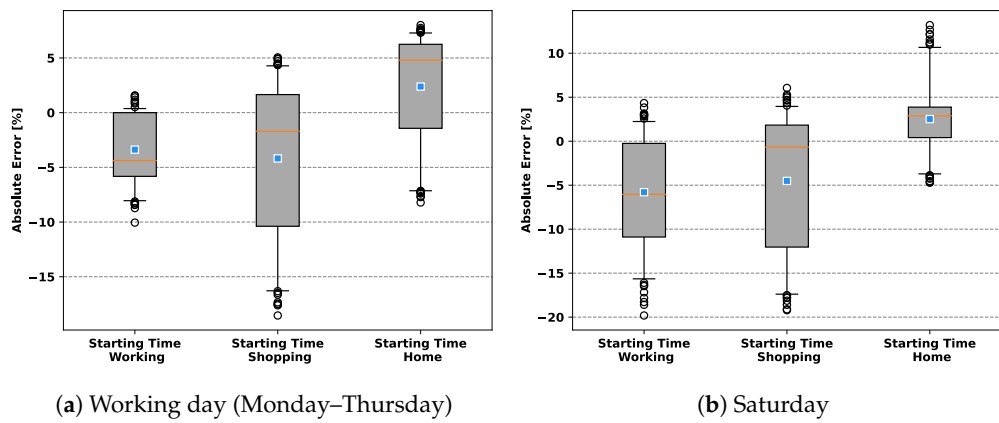

(**a**) Working day (Monday–Thursday)                          (**b**) Saturday

**Figure 10.** Modelling error starting time distributions.

### 3.4. Spatial Distribution of the Charging Energy Demand

By using the mobility model explained in Section 2.2 and the assumptions made in Section 2.3, we calculate the spatial distribution of the charging energy demand in Berlin. Figure 11 shows the resulting spatial distribution of the charging energy demand in the Berlin LORs for a working day (Monday–Thursday). As we consider "home-charging" only, the spatial demand of energy is equally distributed over Berlin on Saturdays compared to working days. The total charging energy demand in Berlin of 4730 MWh on Saturdays is around 14.9% less compared to a working day with 5435 MWh. This can be explained as follows. The average distance of all vehicles is 18.7% higher on working days than on Saturdays. The proportion of moving vehicles is 25.3% higher on working days compared to Saturdays (see Table 3). This leads to the fact that the daily distances of moving vehicles are greater on Saturdays compared to working days. According to Section 2.3, a higher energy consumption is assumed for higher distances. Therefore, the energy consumption on Saturdays is 14.9% lower compared to working days and not, as may expected 18.7% lower. As it can be seen in Figure 11, the energy demand in LORs located in the peripheral areas of the city are higher compared to the ones located in the city center. This is due to a higher degree of motorisation and a higher household income in the peripheral areas. LORs with near zero charging energy demand are pure forest, lake, or industrial areas without inhabitants. It can be seen that LORs with a high energy demand coincide with LORs of high populations, high degrees of motorisation, and high household incomes.

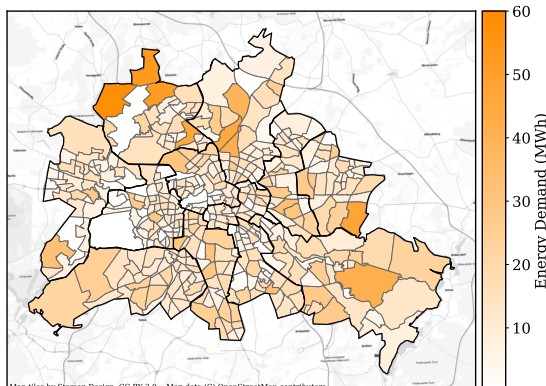

**Figure 11.** Spatial distribution of the battery electric vehicle (BEV) charging energy demand in the Berlin LORs for a working day (Monday–Thursday).

The influence of the factors "degree of motorisation", "population size", and "household income" on the LOR's energy demand is illustrated for five LORs in Table 5. The energy demand is depicted for a working day (Monday–Thursday). The comparison of the

LOR "Stülerstrasse" with the LOR "Huttenkiez" shows the influence of the household income distribution on the energy demand within a LOR. Although more cars are registered in the LOR "Huttenkiez", the LOR "Stülerstrasse" has a higher energy demand (5275 kWh compared to 4441 kWh). This is due to the fact that there are more high-income households in the "Stülerstrasse" LOR, which tend to own larger cars, which require more energy. The influence of the motorisation degree can be studied, for example, by comparing the LOR "Griessingerstrasse" with the LOR "Lübarser Strasse". Both LORs have a similar distribution of household income, with slightly higher income in "Griessingerstrasse". The lower degree of motorisation in "Lübarser Strasse" leads to the lower daily energy consumption of 3423 kWh compared to "Griessingerstrasse", with 6151 kWh.

In Figure 12, the spatial distribution of the BEVs' charging energy demand is depicted per $m^2$ and per inhabitant for the Berlin LORs. It can be seen that the charging energy demand per $m^2$ is higher in the inner city LORs compared to the peripheral areas. The higher population densities within the inner city LORs compensates the lower degree of motorisation and lower household income. The charging energy demand per inhabitant is higher in the peripheral areas of the city, as the degree of motorisation and the household income are higher in the peripheral LORs.

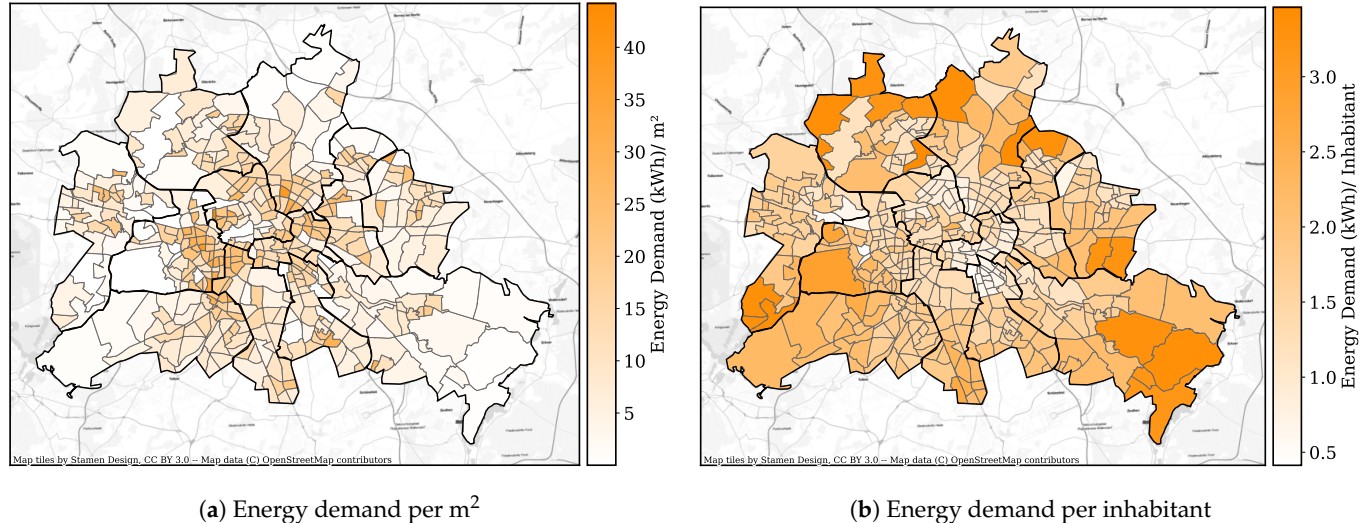

(**a**) Energy demand per $m^2$          (**b**) Energy demand per inhabitant

**Figure 12.** Spatial distribution of the BEV charging energy demand for a working day (Monday–Thursday) normalised to the size and the inhabitants in the Berlin LORs.

**Table 5.** Influence of the household income distribution, the amount of inhabitants, and the amount of vehicles in the LORs on the BEV charging energy demand on a working day (Monday–Thursday).

| LOR | Inhabitants | Vehicles Per 1000 Inhabitants | Household Income Distribution | | | | | Energy Demand (kWh) |
|---|---|---|---|---|---|---|---|---|
| | | | **Very Low** | **Low** | **Medium** | **High** | **Very High** | |
| Stülerstrasse | 3258 | 288 | 0.084 | 0.134 | 0.265 | 0.382 | 0.135 | 5275 |
| Huttenkiez | 3424 | 288 | 0.270 | 0.330 | 0.40 | 0.0 | 0.0 | 4441 |
| Griesingerstr | 3473 | 322 | 0.247 | 0.279 | 0.350 | 0.122 | 0.002 | 6151 |
| Alt-Biesdorf | 3367 | 397 | 0.110 | 0.220 | 0.360 | 0.220 | 0.09 | 7177 |
| Lübarser Strasse | 3214 | 228 | 0.309 | 0.313 | 0.285 | 0.088 | 0.005 | 3423 |

*3.5. Temporal Distribution of the Charging Power Demand*

For charging powers of 3.7 kW and 11 kW, respectively, we evaluate the temporal distribution of the charging power demand for the LOR "Heiligensee". "Heiligensee" is located in the district "Reinickendorf" in the northwest of Berlin. This LOR was chosen for evaluation as it has the highest energy demand of all LORs for the simulated working day. The result is compared to the cumulative power demand of the households within this district. The cumulative power demand of the households is calculated by scaling the standard load profile for Berlin households, which we received from [45]. Standard load profiles are forecasts of the electrical energy consumption at quarter-hourly intervals, provided by the energy supplier. In Germany, they are usually normalised to 1000 kWh per year, which makes scaling necessary [46]. This scaling at the LOR level requires knowledge concerning the distribution of different household sizes as well as annual household electricity consumption. The relevant data is shown in Table 6. We derive the yearly energy demand per household size from [47]. The authors specify the yearly energy demand of different German household sizes as a normal distribution. For our calculations, we use the mean values. The cumulative LOR household power demand is calculated for a working day and a Saturday in May 2018.

In Figure 13a,b, temporal distribution of the charging power demand for a working day (Monday–Thursday) is depicted in blue for 11 kW charging power and 3.7 kW charging power, respectively. In orange, the temporal distribution of the cumulative household power demand within the LOR can be seen. In grey, the superposition with the simulated charging power demand of the BEVs is shown. The results for a Saturday are depicted in Figure 13c for 11 kW charging power and in Figure 13d for 3.7 kW charging power. As it can be seen, the cumulative household power demand between 06:00 and 23:00 is higher on Saturdays, since persons are more likely to be at home, as they do not have to work.

**Table 6.** Amount of households in the LORs "Heiligensee" and "Invalidenstrasse" and their yearly energy demand.

| Household Size | Amount of Households in the LOR "Heiligensee" | Amount of Households in the LOR "Invalidenstrasse" | Yearly Energy Demand Per Household (kWh) |
|---|---|---|---|
| One Person | 4318 | 6894 | 2110 |
| Two Persons | 3046 | 2401 | 3640 |
| Three Persons | 947 | 771 | 4600 |
| Four or more Persons | 1088 | 911 | 4850 |

As depicted in Figure 13, the charging power demand curves of the BEVs for 11 kW charging power show a more pronounced valley and peak compared to their corresponding curves for 3.7 kW charging power. Due to the uncontrolled charging strategy, the vehicles start charging immediately upon arrival at home, which is mainly in the evening hours. The resulting simultaneity of the charging events leads to a peak in the evening hours (maximum around 20:00), which is higher for an 11 kW charging power compared to a 3.7 kW charging power. Since vehicles that are charged with 11 kW charging power need less charging time to fully recharge, the valley of the BEV charging power demand curve is lower for 11 kW charging power compared to 3.7 kW charging power.

As illustrated, the households' power demand peak in the evening coincides with the BEVs' power demand peak for Saturdays and working days. The maximum total power demand of the LOR "Heiligensee" is increased by 77.9% for 11 kW charging power and 59.1% for 3.7 kW on working days with additional BEV load. It is increased by 64.2% for 11 kW charging power and 50.4% for 3.7 kW on Saturdays.

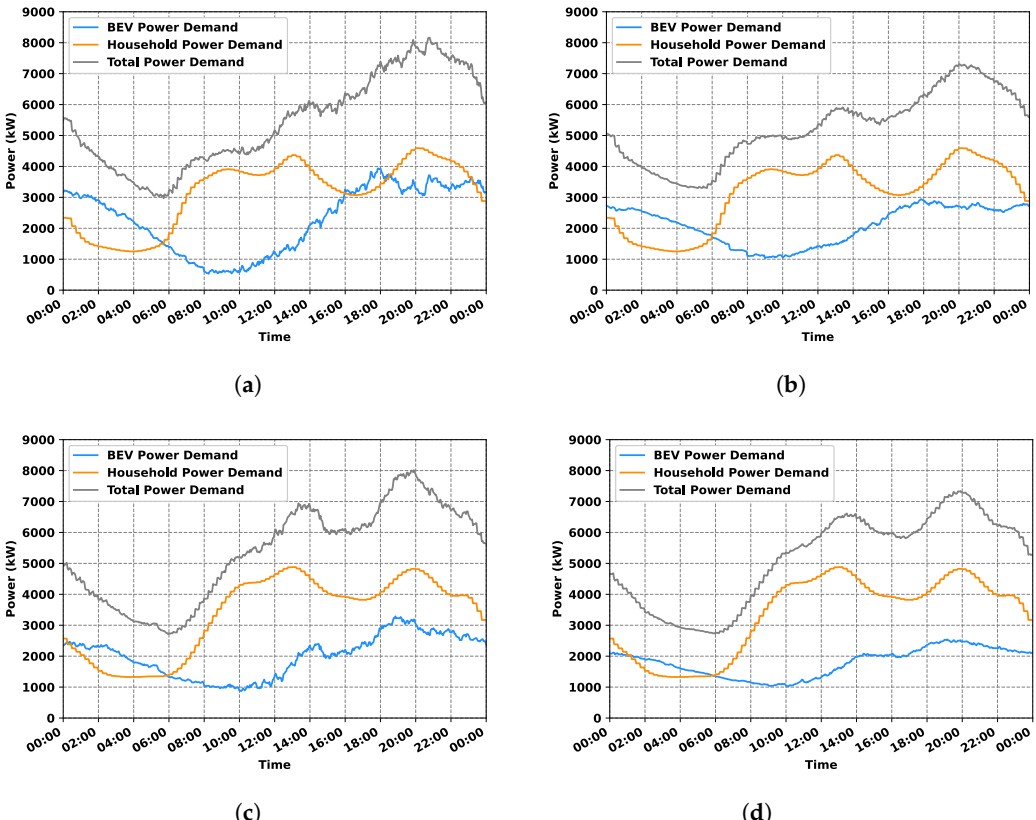

**Figure 13.** Temporal distribution of the BEV charging power demand for the LOR "Heiligensee". (**a**) Working day (Monday–Thursday), 11 kW, (**b**) Working day (Monday–Thursday), 3.7 kW, (**c**) Saturday, 11 kW, (**d**) Saturday, 3.7 kW.

In Figure 14, the temporal distribution of the BEV charging power demand is compared to the cumulative power demand of the households for the LOR "Invalidenstrasse", which is located in the district "Mitte" in the center of Berlin. The results are shown for a working day and charging powers of 3.7 kW and 11 kW. The cumulative household power demand curve is calculated by scaling the standard load profile for Berlin households, as described in the previous paragraph. The distribution of the different household sizes needed for scaling can be found in Table 6. Since more single-person households are located in the LOR "Invalidenstrasse" compared to "Heiligensee", the cumulative household power demand is higher. In comparision to "Heiligensee" the maximum total power demand of the LOR "Invalidenstrasse" is only increased by 19.0% for 11 kW charging power and by 15.7% for 3.7 kW with the additional BEV load. Both LORs have a similar population (18,070 in "Heiligensee" and 17,950 in "Invalidenstrasse") and similar household incomes. Since the degree of motorisation in "Heiligensee" (638 vehicles/1000 inhabitants) is almost four times higher compared to the LOR "Invalidenstrasse" (173 vehicles/1000 inhabitants), the BEV charging power demand is accordingly higher. The temporal distribution of the charging power demand on Saturdays is not depicted for the LOR "Invalidenstrasse", since it shows the same behaviour compared to a working day as already shown for the LOR "Heiligensee".

The computed results of the temporal distribution of the BEV charging power demand can be compared to other studies. The authors of [3] investigated the impact of uncoordinated BEV charging on a medium voltage distribution network. They assumed 1.5 vehicles per household and a 10% electric vehicle penetration rate, resulting in a total of 1270 electric vehicles in the network area. The charging power of the vehicles is between 1.5 kW and 6 kW. They showed that the peak charging power demand can reach values up to 2500 kW in the evening hours. This number is 2.7 times higher than the peak demand

of 900 kW that we determined for the LOR "Invalidenstraße" (3126 BEVs) and 3.7 kW charging power on a working day. The difference arises since the authors assume that all vehicles are used during the simulated day and then are charged almost simultaneously in the evening hours. In contrast to our results, this simultaneity of the charging events reduces the charging power demand in the study to near zero in the early morning.

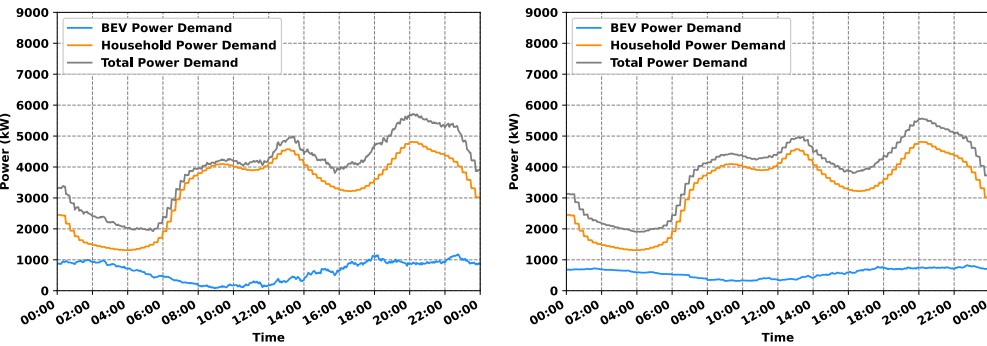

(**a**) Working day (Monday–Thursday), 11 kW   (**b**) Working day (Monday–Thursday), 3.7 kW

**Figure 14.** Temporal distribution of the BEV charging power demand for the LOR "Invalidenstrasse".

The author of [48] investigated the impact of uncoordinated BEV charging on the residential power demand of 200 households in the midwest region of the USA. Similar to our work, the author showed an increased demand in the evening hours. For a motorisation degree of 347 vehicles per 1000 inhabitants, the author showed that the maximum total power demand can increase by 44% for 6.6 kW charging power and 31% for 1.92 kW on working days with the additional BEV load. These findings are comparable to our results. We showed for a charging power of 3.7 kW that the total power demand can increase by 59.1% for a motorisation degree of 638 vehicles per 1000 inhabitants and by 15.7% for a motorisation degree of 173 vehicles per 1000 inhabitants. In contrast to our work, the author showed a similar power demand on Saturdays and on working days, which is mainly due to the different driving behaviours in Berlin and the USA.

## 4. Conclusions and Outlook

In this paper, a methodology was presented to estimate the spatial and temporal distribution of the charging energy and power demand that derives if urban, privately owned cars are fully electrified. The results support the electrical grid operators in detecting and evaluating possible overloads within the electrical grid. The novelty of our approach lies in the development of an activity-based mobility model by directly combining the 2017 German household travel survey (GHTS) with statistics on the population density, the degree of motorisation, and the household income in fine spatial resolution. Through this direct combination, our model provides data-based results with high spatial and temporal resolution. In contrast to other researchers, we take the vehicle class category distribution, which has an significant impact on the energy and power demand, in fine spatial resolution into account. Additionally, unlike most other scientific works, our model is vehicle-based and not person-based. The multiple use of the same vehicle by several persons is therefore better portrayed.

We applied our methodology to the urban area of Berlin in Germany. Since the data sets we used to create our mobility model are available in a similar or identical form to many rural and urban regions in Germany, an upscaling of our model is feasible. Due to our open-data approach, it is relatively easy to obtain similar data sets in most places of the world. Therefore, our methodology is transferable to non-German regions.

The results of our developed mobility model match well with the underlying key data sources. The mean simulation error is between −5.3% and 3.1% for the trip starting time of

the vehicles and the activities "Home", "Working", and "Shopping". For those activities and the trip distance, the mean simulation error is between $-6.1\%$ and $3.3\%$.

The application of our model showed that the spatial distribution of the charging energy demand in Berlin is highly heterogeneous. In the peripheral areas of the city, the total energy demand and the energy demand per capita is higher compared to the city center, mainly because the residents own more cars per capita. Since their household income is higher, the residents in peripheral areas own larger cars that consume more energy. The energy demand per $m^2$ area is higher in the inner city LORs compared to the peripheral areas, since the higher population densities compensate the lower degree of motorisation within the inner city LORs. We additionally showed that the total daily charging demand in Berlin is about 14.9% less on Saturdays compared to working days, which is mainly due to the fact that fewer cars travel on Saturdays.

Regarding the temporal distribution of charging power demand, we compared the cumulative household power demand with the additional BEV charging demand for two LORs. We showed the variability of the BEV charging power demand in the LORs. While the total power demand is only increased by 19.0% in the LOR "Invalidenstraße" on a working day and a charging power of 11 kW, it is increased by up to 77.9% in the LOR "Heiligensee". For a charging power of 3.7 kW, the increase is 59.1% in the LOR "Heiligensee". These results shows the necessity for and are the basis for intelligent load shifting methods. Those methods can be used to reduce the charging power demand peaks and to flatten the load curve, as shown by other studies [4,7,49–51].

In further research, we will investigate load-shifting potentials by combining our model with a smart charging algorithm. We will investigate if the BEVs' charging power demand peaks can be reduced without limiting the vehicle's range. Furthermore, as we currently solely consider "home-charging" in our model, we will use geodata to analyse the building structure in each LOR. The results can be used to evaluate the attractiveness for vehicles to enter the LORs for specific activities. By using a routing algorithm and the attractiveness information, the daily routes for all vehicles can be determined. The resulting knowledge about the location and the duration of stay for the different activities will be used for the investigation of combined charging strategies.

**Author Contributions:** Conceptualisation and supervision, F.S.; investigation and methodology, F.S. and S.S.; project administration and funding acquisition, D.G.; validation and visualisation, F.S.; writing—original draft, F.S.; writing—review and editing, F.S. and D.G. All authors have read and agreed to the published version of the manuscript.

**Funding:** This research was funded by the Deutsche Forschungsgemeinschaft (DFG, German Research Foundation)—project "moreEVS", grant number: 410830482.

**Institutional Review Board Statement:** Not applicable.

**Informed Consent Statement:** Not applicable.

**Data Availability Statement:** Not applicable.

**Conflicts of Interest:** The authors declare no conflicts of interest.

## Abbreviations

The following abbreviations are used in this paper:

| | |
|---|---|
| BEV | Battery Electric Vehicle |
| ICEV | Internal Combustion Engine Vehicle |
| GHTS | German Household Travel Survey |
| LOR | Lebensweltlich Orientierter Raum |

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
