# Peer review of "Methodology for Estimating the Spatial and Temporal Power Demand of Private Electric Vehicles for an Entire Urban Region Using Open Data"

_energies, doi:10.3390/en14082081_

Round 1

Reviewer 1 Report

The topic of the work is topical and methodically very complex.

Abstract and Introduction

There is a lot of work on charging and forecasting systems. So please show more in the summary and introduction the scientific value of the research undertaken. Which research gap is filled by work. How is this job different from others? Sometimes it is enough to describe it better.

It would be useful to have a broader review of the literature at the beginning of the work on the issues of this type of research

The results are correct. The presentation and form of the research conducted are good.

Present the results in the form of points. This will facilitate the analysis of work achievements. Work is valuable and it must be agreed that the results from real studies are much more valuable than model studies. However, they cover one area of Berlin. It is worth to generalize in the form of conclusions how the conducted research can be used on a larger scale.

Reviewer 2 Report

This manuscript has used available information to look at the impact of electric vehicles on the electrical grid.  The assumptions on page 9 limit the usefulness of the results.  The time of charging can be managed better with time of use rates.  The management of the grid and EV charging is better if the charging takes place at the best locations for the owner.  Long trips were eliminated because of the limitation of home charging, and this distorts the results.

The data on number of vehicles on page 4, line 162 is too large and not in agreement with other data in the paper.

No consideration is given to the impact of electric buses.

There is a need to make use of the smart grid and modern communication to manage peak power issues.

The manuscript is very long in its present form.  Information not related to the topic should be reduced or eliminated.  See lines 206-215 on page 5 for example.

The charging rate of 11kW is very large for the average vehicle that travels the average distance.  The results in Figure 13a are not what one would expect for the data that are presented elsewhere in the manuscript.  
